# Forest Ecosystem Services-Based Adaptation Actions Supported by the National Policy on Climate Change for Namibia: Effectiveness, Indicators, and Challenges

Andreas Nikodemus *, Miroslav Hájek, Albertina Ndeinoma and Ratna Chrismiari Purwestri

Faculty of Forestry and Wood Sciences, Czech University of Life Sciences Prague, Kamýcká 129, 165 00 Prague, Czech Republic
* Correspondence: nikodemus@fld.czu.cz

**Abstract:** Forest ecosystem services are crucial in adaptation, mitigation, and increasing climate change resilience. Although most climate change policies promote adaptation actions in forest ecosystem services, there are limited studies focusing on the forest ecosystem services-based adaptation actions supported by the National Policy on Climate Change for Namibia (NPCC). This paper aims to assess the effectiveness of forestry adaptation actions of the NPCC. An independent *t*-test for non-categorical data was used for the statistical analysis to compare mean scores of the implementation effectiveness of adaptation actions and challenges before and after the NPCC implementation, according to the perceptions of forestry and climate change cross-sectoral experts. A *p*-value less than 0.05 ($p < 0.05$) was designated as the statistical significance. Adaptation actions in forest ecosystem services were significantly effective after the introduction of the NPCC. Biodiversity and carbon sequestration were significantly effective after the introduction of the NPCC. The most significant challenges identified were the lack of awareness, which affected adaptation actions before and after the policy. Afforestation, reforestation, awareness, and forestry research need strengthening to improve the effectiveness of the NPCC. Although our results showed that adaptation actions supported by the NPCC were generally effective after the introduction of the policy, we identified some implementation areas that require strengthening, mainly through research, to help in sound decision-making. We, therefore, recommend future research to analyze the strengths, weaknesses, threats, and opportunities (SWOT) of the NPCC and consequently design/propose a framework for forest ecosystem services-based adaptation actions in the policy to improve adaptation actions.

**Keywords:** biodiversity; carbon sequestration; soil conservation; socio-economic benefits; Southern Africa; local communities; vulnerability

## 1. Introduction

All actions toward climate change adaptation at all levels comply with the Paris Agreement, which aims to reach an international goal of adaptation to climate change [1,2]. This goal seeks to ensure an adequate adaptation response to the global temperature goal, enhancing adaptive capacity, strengthening resilience, and reducing vulnerability to climate change, ultimately contributing to sustainable development [2]. Hence, achieving an adequate adaptation response to the impacts of climate change will require continuous efforts from integrated policy instruments at global, regional, and national levels [3].

On the national level, most countries formulated cross-sectoral policy instruments and strategic national-level actions to promote climate-friendly forestry activities while discouraging climate-adverse ones [4]. For example, China adopted a low-carbon city pilot policy, which was evaluated to effectively reduce carbon emissions while negatively affecting urban land use efficiency [5]. One of the main actions is restoring the vulnerable forests to regain vitality and vigor while safeguarding the local livelihood options [6].

Carbon sequestration, watershed services, soil conservation, biodiversity, and recreational and cultural values [6,7] are part of the primary forest ecosystem services that play a critical role in climate change adaptation and mitigation [8,9]. Climate change affects these forest ecosystem services differently [10]. For example, climate change has a direct and indirect influence on forest biodiversity across the globe [11]. As a result, political support is essential for the systematic integration of ecosystem management into climate change adaptation and policy frameworks and practices [12]. In addition, ecosystem-based climate change adaptation is now recognized by international agreements and policy instruments [13].

The implementation of climate change adaptation policies can be nature-based or technical. However, in the context of European forest ecosystems, nature-based policies, including biodiversity, ecosystem services, and human well-being, were more cost-effective and better at coping with the ethical and inequality issues associated with the distributional impacts of the policy actions [14]. Although ecosystem-based policies might differ in terms of the ecosystem services they focus on, they must be coherent [15].

Namibia is the driest country in sub-Saharan Africa [16]. The country is characterized by high climatic variability in the form of persistent droughts, unpredictable and variable rainfall patterns, variability in temperatures, and scarcity of water [17,18]. The climate in Namibia is typically hot and dry, with an average annual temperature of 18–22 °C [19]. As a result, the country is significantly vulnerable to the impacts of climate change [20,21]. In addition to its highly variable climate, Namibia's acute vulnerability to climate change is also influenced by the high reliance of local livelihoods and important economic sectors on climate-related natural resources such as forest ecosystem services [22–26].

The unique climate conditions of Namibia and its high vulnerability to climate change call for robust policies to guide action on climate change in Namibia at the national level [18]. Hence, the NPCC was adopted in 2011 [27]. The National Climate Change Committee (NCCC) oversees the implementation of the NPCC and comprises representatives of various ministries and other stakeholders, such as the private sector and NGOs [27,28]. The NCCC is chaired by the Ministry of Environment, Forestry, and Tourism (MEFT). The NPCC provides an institutional framework and overarching national strategy for developing, implementing, monitoring, and evaluating climate change mitigation and adaptation activities in Namibia [28]. The NPCC aims to lower Namibia's vulnerability to climate change to contribute to sustainable development in line with Namibia's Vision 2030 [29].

Since climate change is a complex global problem [30], the NPCC was designed to manage climate change responses in a way that recognizes national developmental goals and promotes the integration and coordination of programs of various sector organizations [16]. While climate change issues have been mainstreamed across the country's key sectors, such as agriculture, water resources, tourism, and health, these policies do not include concrete actions to mitigate climate change risks [27]. Hence, Namibia is currently developing its first Nationally Appropriate Mitigation Action (NAMA) and is working on its National Adaptation Plan (NAP) to better guide the country on its way to mitigating and adapting to climate change [27].

Forest ecosystem services play a crucial role in adaptation, mitigation, and increasing resilience to climate change [31]. Forest ecosystem services such as carbon sequestration and biodiversity all contribute to climate change adaptation [13]. Due to its dry conditions, temperature variability, and erratic rainfalls [17,18], Namibia's forests are characterized by savannah woodlands with a combination of trees and shrubs [32]. Despite the status of the forests of Namibia, the question that remains not answered is whether adaptation actions supported by the NPCC were framed in such a way that they promote resilience, adaptation, and mitigation in the context of forest ecosystem services at the national level.

In the context of forest ecosystem services, the focus area of the NPCC encompasses afforestation, reforestation, agroforestry, commercial forestry, community-based forest management, and woodland management [16]. Although continued efforts to increase the country's resilience capabilities and strengthen the country's social and economic

structures against vulnerability take forestry into account as one of the country's most vulnerable sectors [27], there is limited scientific knowledge about specific adaptation actions focusing on Namibia's unique forest ecosystem services within the framework of the NPCC. Secondly, there is no clear scientific evidence of the effectiveness and challenges facing the existing adaptation actions supported by the NPCC in forest ecosystem services. Furthermore, the factors influencing the implementation of the NPCC adaptation actions in forestry have not yet been investigated.

Therefore, it is unclear whether the existing NPCC's measures for climate change adaptation in forest ecosystem services at the national level are adequate. Hence, it is difficult for policymakers to formulate policy actions that address climate change adaptation adequately through forest ecosystem services. Thus, it is crucial to establish a sound understanding in this area because when forest ecosystem managers and policymakers are well-informed, they can benefit from policy actions to support climate change mitigation and adaptation actions [33].

This paper aims to assess the effectiveness of forest ecosystem services-based adaptation actions supported by the NPCC. To achieve the paper's goal, we compare the current adaptation actions with the measures that were implemented before the policy's introduction. Finally, we propose improvements to effectively implement the NPCC and strengthen the adaptive capacity of all types of forest ecosystem services in Namibia.

## 2. Methods

### 2.1. Study Area

The study focused on Namibia, a developing country situated in south-western Africa, between latitude 17° S and 29° S and longitude 11° E and 26° E. It shares borders with Angola to the north, South Africa to the south, Botswana to the east, and Zambia to the northeast [19,27]. Namibia is a sparsely populated country with a population of 2.5 million and covers a total surface area of 824,292 km$^2$ [34].

In addition, its dry conditions significantly influence forest cover [26,35]. It is estimated that forests and woodlands in Namibia cover approximately 20% (about 53 million ha) of the total surface area [36]. Various factors, such as land use, including crop cultivation, affect forest cover in Namibia. In addition, vegetation types are distributed across the country according to climate variability (Figure 1).

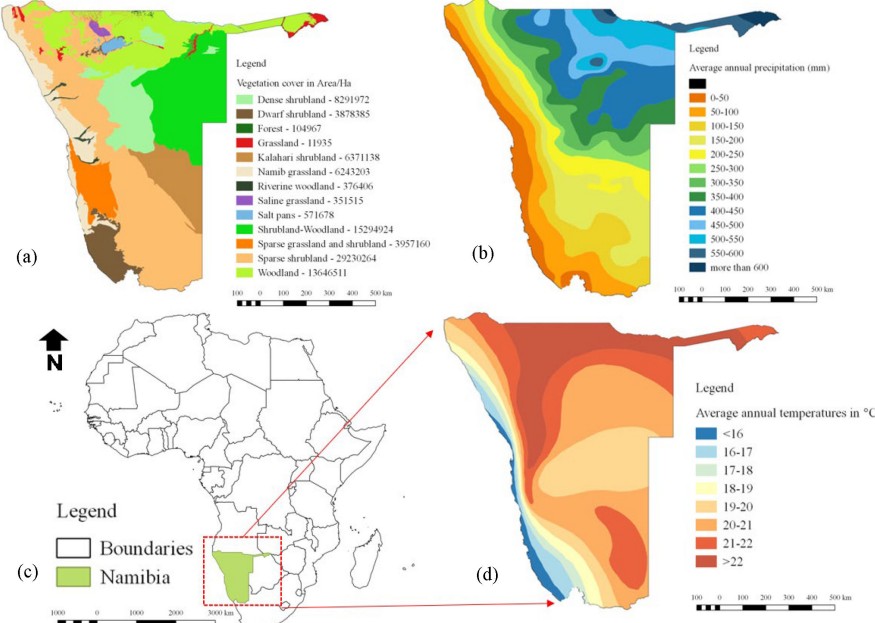

**Figure 1.** A map of the study area (Namibia) and key descriptions, (**a**) vegetation types, (**b**) precipitation variability, (**c**) the location of Namibia on the map of Africa, and (**d**) temperature variability.

Namibia is between two deserts; the Namib Desert stretches along its west coast, and the Kalahari Desert borders its eastern and southern neighbors, Botswana and South Africa [26]. Due to its geographical location, Namibia's three main vegetation types can be classified as woodlands, savannas (grass cover, trees, and shrubs), and deserts (Namib grassland) [37,38]. Therefore, it is worth noting that climate variability and the nature of vegetation types are the main attributes influencing the level of adaptation actions in forest ecosystems in Namibia.

*2.2. Survey*

To achieve the aim of the study, we purposively collected data from forestry and climate change experts representing different institutions, including public, private, and government projects, academics, and researchers. We selected specific institutions based on their involvement in climate change adaptation and related activities, mainly research and forest ecosystems management. Since we focused on the practical implementation of the policy, which required a deeper understanding of the policy, we excluded ordinary citizens. Ordinary citizens lack practical understanding of the implementation of policy instruments for climate change adaptation actions [39].

From public institutions, we focused on senior employees, for example, in the Directorate of Forestry (DoF), which is the custodian of forest ecosystem services. We also involved senior employees from the Ministry of Agriculture, Water and Land Reform (MAWLR). Agriculture and forestry have various integrated management approaches that influence forest management practices in Namibia. We also included multiple projects under the Climate Unit of MEFT, such as the Namibia Integrated Landscape Approach for Enhancing Livelihoods and Environmental Governance to Eradicate Poverty (NILALEG), Deutsche Gesellschaft für Internationale Zusammenarbeit (GIZ), the National Communications, Biannual update reports, and Greenhouse gas inventory, Capacity-building Initiative for Transparency Stakeholder engagement (CBIT) and Climate Promise and the Southern African Science Service Centre for Climate Change and Adaptive Land Management (SASSCAL).

For academic institutions, we focused on the lecturers and researchers from the two prominent local universities, namely the departments of environmental sciences at the University of Namibia (UNAM) and the department of agriculture and natural resource sciences at the Namibia University of Science and Technology (NUST).

Since we assessed the effectiveness of adaptation actions before (2001–2010) and after the policy's launch (2011–2021), there were few experts with relevant experience in implementing climate change adaptation approaches in forestry. As mentioned earlier, Namibia is a sparsely populated country with sparsely forested land. As a result, we purposively shared the survey link with 40 cross-sectional experts. However, we could only collect results from 36 cross-sectoral experts, translating into a 90% response rate.

*2.3. Data Collection*

We administered the questionnaire (Appendix A) to the experts from 27 August 2022 to 30 September 2022, which accounts for 35 days of data collection, including weekends and public holidays. We shared the link for an online semi-structured questionnaire (Survio 2022 version) with target respondents via email, WhatsApp, and LinkedIn. We used these platforms because they are user-friendly, cheap, and commonly used by most professionals daily. We employed an online questionnaire due to its attributes, such that it is less costly, less time-consuming, flexible, and convenient to complete, especially for senior experts occupying busy offices.

Since most of the experts hold higher positions with busy schedules, we made several follow-ups to remind them to participate in the survey. We strategically sent reminder alerts every Monday and every Friday of the week during the survey period. A pre-test survey was conducted with two respondents to ensure the relevancy and accuracy of the questions before the actual data collection.

### 2.4. Statistical Analysis

We used the independent *t*-test for non-categorical data for the statistical analysis to compare mean scores of the implementation effectiveness, actions, and challenges before and after the NPCC implementation, according to the expert's perceptions (Appendix B). We designated a *p*-value less than 0.05 ($p < 0.05$) as the statistical significance. We performed all the analysis using IBM SPSS version 28 (IBM Corp. Armonk, NY, USA).

The independent samples *t*-test can be represented using the functions below:

$$t = \frac{\overline{x}_1 - \overline{x}_2}{s_p \sqrt{\frac{1}{n_1} + \frac{1}{n_2}}}$$

with

$$s_p = \sqrt{\frac{(n_1 - 1)s_1^2 + (n_2 - 1)s_2^2}{n_1 + n_2 - 2}}$$

where

$\overline{x}_1$ = Mean of first sample
$\overline{x}_2$ = Mean of second sample
$n_1$ = Sample size (i.e., number of observations) of first sample
$n_2$ = Sample size (i.e., number of observations) of second sample
$s_1$ = Standard deviation of first sample
$s_2$ = Standard deviation of second sample
$s_p$ = Pooled standard deviation

As mentioned earlier, our analysis focused on 10 time series before and after the introduction of the NPCC. That is, 10 years (2001–2010) before the policy's launch and 10 years after (2012–2021). We excluded the year 2011 because the policy's effects were most likely not evident in the first year of its implementation. Second, to establish the impact of the temporal implementation status of the policy on the adaptive capacity of forest ecosystem services, we computed an independent *t*-test to compare the overall adaptation levels.

### 2.5. Qualitative Analysis

For qualitative analysis, we used ATLAS.ti Scientific Software Development GmbH version 22.2.4 (Berlin, Germany) to code and organize qualitative data. Qualitative data were used to explore the experts' perceptions about possible improvements for the implementation actions of the NPCC. Qualitative data were coded according to relevant themes (codes) derived from the proposed revisions for the NPCC.

## 3. Results

### 3.1. Effectiveness

Forest ecosystem services manifest primarily in seven services (Table 1). Since climate change affects each type of forest ecosystem in different ways [10], our assessments for the effectiveness of adaptation actions were based on the main forest ecosystems, namely biodiversity, carbon sequestration, soil conservation, socio-economic benefits, recreational and cultural values, watershed services, and high conservation values.

Our results showed that adaptation actions in forest ecosystem services, namely biodiversity (4.36 ± 1.52), carbon sequestration (3.06 ± 1.35), soil conservation (3.39 ± 1.29), and socio-economic benefits (3.44 ± 1.34), were more effective after the NPC. Notably, adaptation actions were significantly higher in biodiversity than in the rest of the forest ecosystem services. In other words, biodiversity's mean effectiveness score after NPCC (4.36 ± 1.52) was significantly higher than before (3.11 ± 0.92) ($p < 0.001$). Similarly, the mean effectiveness score in carbon sequestration was also higher after NPCC (3.06 ± 1.35) than before (2.75 ± 1.34). Although the rest of the forest ecosystem services are not

statistically significantly different, it can be said that adaptation actions after NPCC were more effective than before.

**Table 1.** Effectiveness scores of the implemented adaptation actions by forest ecosystem services before and after NPCC [1].

| Forest Ecosystem Services | Before NPCC (N = 36) | After NPCC (N = 36) | *p*-Value [2] |
|---|---|---|---|
| Biodiversity | 3.11 ± 0.92 | 4.36 ± 1.52 | <0.001 |
| Carbon sequestration | 2.75 ± 1.34 | 3.06 ± 1.35 | 0.338 |
| Soil conservation | 3.08 ± 1.32 | 3.39 ± 1.29 | 0.324 |
| Socio-economic benefits | 3.25 ± 1.20 | 3.44 ± 1.34 | 0.519 |
| Recreational and cultural values | 3.72 ± 1.09 | 3.39 ± 1.34 | 0.249 |
| Watershed services | 3.31 ± 0.89 | 3.17 ± 1.23 | 0.585 |
| High conservation values | 3.56 ± 1.40 | 3.14 ± 1.25 | 0.187 |

[1] Data are presented as mean ± standard deviation (sd); independent *t*-test was applied to compare mean scores before and after NPCC implementation. [2] Significantly different at $p < 0.05$.

### 3.2. Adaptation Actions Indicators

We established indicators for adaptation actions to assess the effectiveness of forest ecosystems before and after the NPCC (Table 2). Our assessments focused on the main adaptation action indicators supported by the policy.

**Table 2.** Indicators of actions scores of the implemented policy before and after NPCC [1].

| Adaptation Action Indicators | Before NPCC (N = 36) | After NPCC (N = 36) | *p*-Value [2] |
|---|---|---|---|
| Afforestation and reforestation | 3.17 ± 1.56 | 3.33 ± 1.69 | 0.665 |
| Law enforcement | 3.33 ± 1.51 | 3.61 ± 1.63 | 0.455 |
| Altering local communities' reliance on forest resources | 3.42 ± 1.44 | 3.56 ± 1.59 | 0.699 |
| Funding adaptation activities | 3.11 ± 1.58 | 3.53 ± 1.42 | 0.244 |
| Forestry research | 3.17 ± 1.63 | 3.28 ± 1.60 | 0.771 |
| Conservation of ecosystem services critically threatened by climate change | 30.6 ± 1.41 | 3.53 ± 1.42 | 0.162 |
| Stakeholders' collaboration | 2.94 ± 1.64 | 3.53 ± 1.40 | 0.109 |

[1] Data are presented as mean ± sd; independent *t*-test was applied to compare mean scores before and after NPCC implementation. [2] Significantly different at $p < 0.05$.

There was no statistically significant difference among the indicators of adaptation actions before and after the NPCC. However, all the adaptation actions showed higher mean effectiveness scores after the NPCC. Law enforcement (3.61 ± 1.63) and altering local communities' reliance on forest resources (3.56 ± 1.59) were the most effective adaptation action indicators after the introduction of the NPCC. Conversely, afforestation and reforestation (3.33 ± 1.69) and forestry research (3.28 ± 1.60) were also effective after the NPCC's launch. However, these two adaptation actions showed the lowest effectiveness scores after the NPCC.

### 3.3. Challenges

There are several challenges facing implementing the adaptation actions to climate change supported by the NPCC in forest ecosystem services. In this regard, our assessments focused on the main challenges, such as lack of awareness, high demands for agricultural land, limited research, adverse weather conditions, poverty in rural areas, lack of funding options, and poor stakeholders' collaboration (Figure 2).

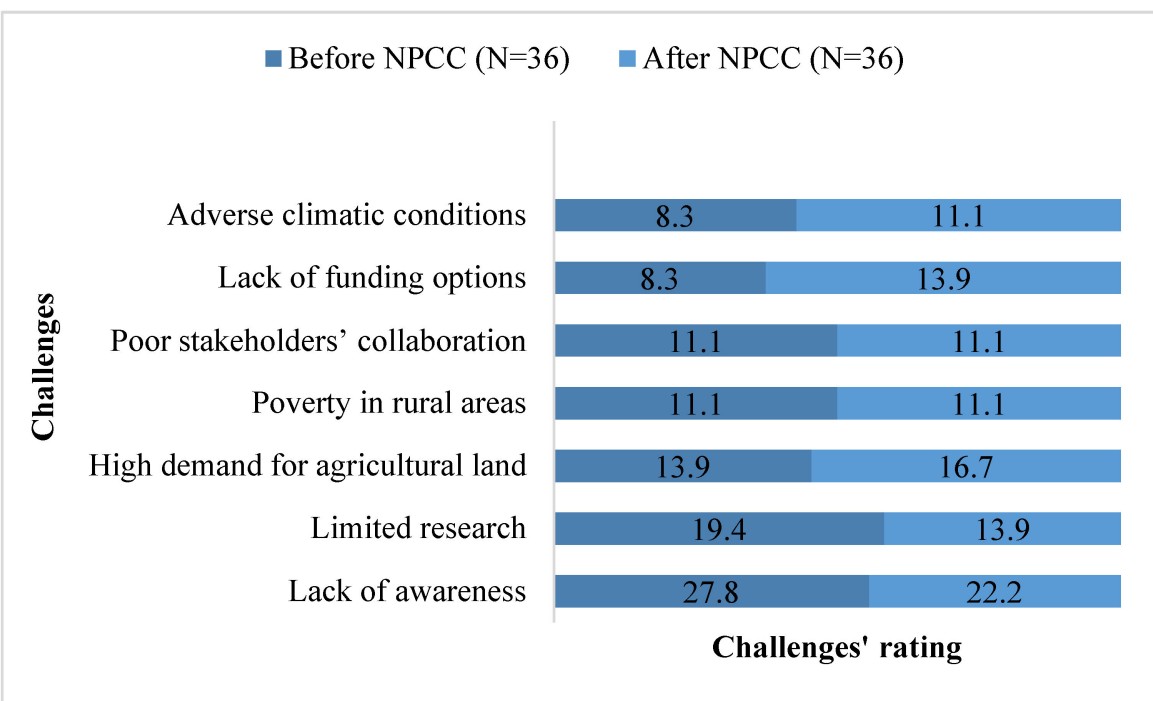

**Figure 2.** Challenges in the implementation of adaptation actions before and after the NPCC. Challenges' ratings are presented as %.

The most significant challenges were the lack of awareness (27.8% before the NPCC; 22.2% after the NPCC). The lack of awareness in this context refers to the limited information and general understanding of climate change and its impacts in the context of forest ecosystem services. This is one of the obstacles; it affects forest ecosystems and the implementation of adaptation actions [39]. Another severe challenge was limited research (in forestry), rated 19.4% before the NPCC and 13.9% after the NPCC.

The most significant challenges were highly significant before the NPCC. However, challenges such as high demand for agricultural land (16.7%), the lack of funding options (13.9%), and adverse weather conditions (11.1%) were significant after the NPCC.

Overall, it can be said that most challenges facing adaptation actions supported by the NPCC were more severe after the introduction of the NPCC. This situation could be attributed to various factors, including land use changes and management practices. However, research has yet to establish scientific evidence on this aspect.

*3.4. Proposed Improvements*

According to the experts, there are multiple areas of adaptation actions in forest ecosystem services that need enhancement to improve the effectiveness of the NPCC (Figure 3).

The experts expressed that promoting awareness (33.3%) was the most critical improvement needed to improve the effectiveness of the NPCC. Creating awareness is crucial in promoting adaptation actions because local knowledge is vital to help local communities cope with climate change and variability. Furthermore, awareness creation catalyzes sustainable forest ecosystem management [40].

Additionally, experts further indicated a need for strengthening forestry and climate change research (13.9%). Experts further pointed out that enhancing adaptation measures (11.1%) and availing sufficient funds (11.1%) are other areas that need improvements to increase the effectiveness of the NPCC. The experts also listed promoting the carbon market (2.8%) and renewable energy (2.8%) among the proposed improvements, but with the lowest significance level.

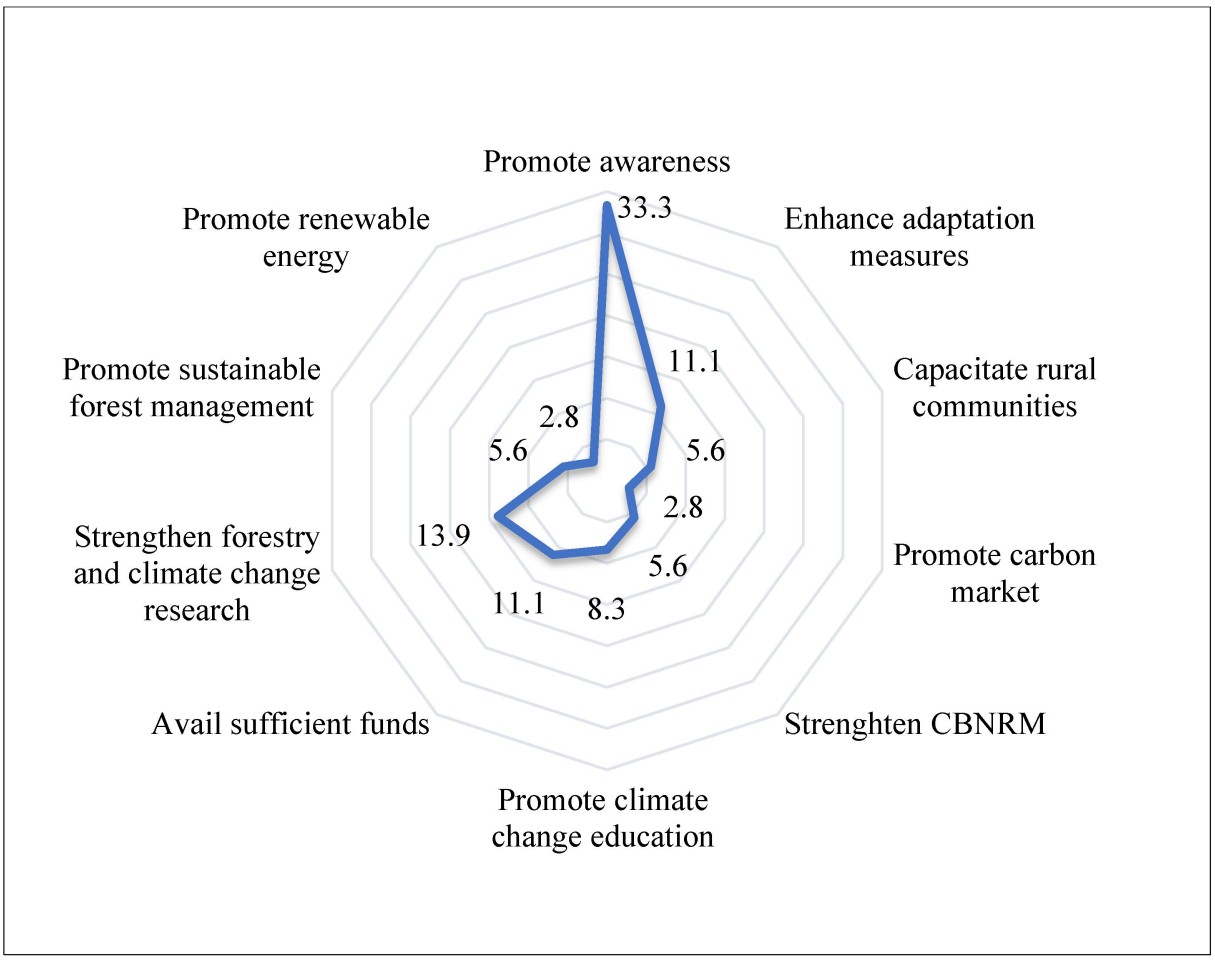

**Figure 3.** Proposed improvements in the implementation of the NPCC, according to the experts. Proposed improvements for the NPCC are presented as %.

## 4. Discussion

Forest ecosystems are crucial for adaptation to climate change. Despite forest ecosystems' vast ecological and livelihood importance, they are highly threatened by global changes [41–43]. Therefore, countries have taken different approaches to integrate climate change adaptation into their environmental laws and policies [44,45]. There is a need to incorporate climate change conditions in decision-making and policy formulation to maintain ecosystem capacity across different sectors and social statuses, including rural and urban areas [46]. However, we noted that most forest ecosystem-based policies could be broad in most countries. For example, in India, forest policies have been broadly aimed at conservation, reducing pressure on forests, and providing biomass to the large forest-dependent population for their fuel and fodder needs, apart from generating revenue through the production and sale of timber [47]. Therefore, since climate change significantly impacts forest ecosystems [9,47,48], there is a dire need to revisit forest ecosystem-based policies in the context of climate change adaptation actions, their effectiveness, challenges, and opportunities, and the national level in various countries across the globe.

Furthermore, since climate change is a global phenomenon, it is encouraged that linking local efforts with international initiatives is likely to produce more significant results [49]. One international approach to responding to climate change's effects on forests is forest genetic modification. In this view, since genetic diversity is a crucial component of resilience and adaptability [50], countries are encouraged to include genetic-level responses to climate change in their action plans [51]. However, research gaps remain in this aspect of forest ecosystems and climate change adaptation.

Namibia is among the few countries that emphasize implementing climate change policies at the national level. Although most countries such as Zambia, Mali, and Tanzania implement forest ecosystems in community-based coping strategies [52], national climate change policies exist and mainly emphasize cross-sectoral adaptation actions, including ecosystems' integrity. Although national climate change policy actions in most counties consider the critical role of ecosystems in reducing forest degradation and loss of forest ecosystems [53,54], there seems to be little emphasis on their effectiveness in this regard from the research perspective.

In the case of Namibia, implementing the climate change policy at the national level is crucial, considering that Namibia is the driest country in Sub-Saharan Africa and, hence, one of the most vulnerable countries to the effects of climate change [21]. This situation exposes the country's forest ecosystem services to the severe impacts of climate change. As a result, implementing adaptation actions for climate change is critical [55], especially in forest ecosystem services. However, to ensure effectiveness in adaptation actions, it is essential to implement robust policy instruments.

In this paper, our assessments focused on implementing the NPCC to support adaptation to climate change in forest ecosystem services. Although it is difficult to compare the effectiveness of local policies due to differences in forest ecosystem conditions, the existing literature shows that climate change adaptations at the policy level are insufficiently mainstreamed within broader development approaches in the forest ecosystems context [56].

The goal of the NPCC is to manage climate change responses on the national level [57]. Based on our results, it is evident that the implementation of the NPCC has played a significant role in supporting adaptation actions in forest ecosystem services in Namibia. The effectiveness of the policy was significantly manifested in biodiversity ($p < 0.001$) (Table 1). Although no previous studies provide evidence of the status of biodiversity after the policy was introduced, our results indicate that biodiversity's role as a remedy to climate change has improved through the NPCC. Another function of forest ecosystems is to provide habitats for biodiversity [58]. Forest biodiversity also plays a critical role in carbon sequestration. Our results indicated that carbon sequestration also proved effective after the NPCC. Carbon is stored in five distinct pools in forest ecosystems, namely, above-ground and below-ground live biomass, in deadwood, including snags, litter, and soil [58]. In that way, forest ecosystems' biodiversity plays a critical role in promoting adaptation and resilience to the impacts of climate change. Furthermore, we also identified soil conservation, socio-economic benefits, and recreational and cultural values among the primary forest ecosystem services in which the effectiveness of the policy was significant.

Management policies will more strongly determine the future provision of forest ecosystem services [59]. However, our results noted that the effectiveness of the adaptation actions was not significantly different before and after the policy (NPCC) was introduced in 2011. This situation could be attributed to factors such as the absence of changes in management approaches for forest ecosystem services. In addition to the attitude and behavior of local communities, forest management practices also influence adaptation actions significantly [60]. Furthermore, we noted that another factor that could have influenced the effectiveness of the policy is potentially the fact that it is still in its infancy stage (10 years) of implementation.

Our results are unique in that we focused specifically on the performance of the NPCC in forest ecosystem services at the national level. However, our results suggest that substantial research gaps exist in the context of climate change and forest ecosystem services in Namibia and many other countries around the globe. Most existing studies in different parts of the world focused on the policy guidelines [61] instead of their practical implementation. In South Africa, for example, current research focused on the policy-making process [62] and not necessarily its implementation, particularly in forest ecosystem services.

All adaptation actions supported by the NPCC align with the mission statements of MEFT, which hosts the NCCC [28]. The NCCC oversees the implementation of the NPCC.

Although there was no statistically significant difference among the indicators of adaptation actions before and after the NPCC, our results revealed that all the adaptation actions in our assessments showed higher mean effectiveness scores after the NPCC. Law enforcement and altering local communities' reliance on forest resources (Table 2) were the most effective adaptation action indicators after the introduction of the NPCC. Despite several obstacles that need to be addressed (Figure 2), our overall results proved that the policy effectively supports the existing adaptation actions in various forest ecosystem services.

Even though the policy proved effective in promoting adaptation actions, our results revealed several challenges facing the effectiveness of the policy (Figure 2). The most significant challenges affecting the effectiveness of the policy were the lack of awareness and limited research on forest ecosystem services and climate change. In addition, the high demand for agricultural land and the lack of funding options also affects the policy's implementation. Therefore, our results agree that the design and implementation of climate policies for forest ecosystem-based services should respect the country-specific environmental, economic, and political contexts [63,64].

Additionally, our results noted that implementing climate change policy alone is not enough. Sustainable funds should support it. However, it is worth highlighting that ecosystem-based adaptation actions are costly [65]. The lack of funds for adaptation is an issue in many developing countries, particularly in Africa [66]. For example, South Africa established that improving resources, including funding, was listed among the areas that need strengthening to enhance adaptive capacity [67]. In the same view, regarding adaptation, ecosystems, including forest ecosystems, were listed among the substantially underfunded areas in Africa [68].

The existing global funds seem ineffective in their intended approaches to finance adaptation to climate change. One of the worldwide climate change adaptation funds is the United Nations Framework Convention on Climate Change's (UNFCCC) Green Climate Fund (GCF), which is a financial mechanism designed to fund adaptation actions [69]. The fund has pledged to promote local adaptation funding in underdeveloped nations. However, it has not successfully operationalized this pledge [69]. Hence, it has been established that countries, especially developing ones, including Namibia, require support for implementing and diffusing prioritized technologies, mainly in the energy, agriculture, forestry, and other land use and water sectors [3].

Our results revealed a lack of awareness about climate change and its impacts on forest ecosystem services (Figure 3). It is worth noting that understanding how the climate affects forests, industries, and local communities and how these effects can evolve and incorporating this knowledge into management decisions are all necessary for adaptation actions and climate change policies [70]. Generally, the lack of awareness is an issue among the local rural communities who live in proximity to forest resources in Namibia [39]. This situation challenges the sustainable management of forest resources and consequently contributes to the impacts of climate change on the national level. Therefore, it is crucial to prioritize and avail information and tools to make decisions in solving climate change's effect on forest ecosystems [31]. This goal can be achieved through research about climate change and forestry, which is one of the areas that needs urgent attention in the context of climate change and forestry in Namibia.

Another area that needs improvements is capacity building in rural communities (Figure 3). According to the mission statement of DoF, local communities are mandated to have access to forest resources and utilize them sustainably through the community forest project [24]. However, this approach requires stable funding mechanisms to monitor and ensure local communities' sustainable use of forest resources. The weakness displayed in Community-Based Natural Resources Management (CBNRM) is common in African countries [71]. Therefore, it is vital to avail funds for forest management practices through CBNRM to maximize monitoring during the establishment of participatory forest management associations and maximize its contribution to climate change adaptation.

The results entail that efforts to enhance the effectiveness of adaptation actions of the NPCC to climate change in forest ecosystem services must include steps taken to strengthen climate awareness and understanding amongst forest managers, climate change scientists, local communities, and policymakers. Therefore, approaches such as robust research and continuously engaging all stakeholders in climate discourse, capacity building, and tailor-made climate and forest ecosystems will need to be incorporated. To achieve this, the government and stakeholders should transform the policy into a mainstreaming forest ecosystem-based adaptation policy that applies in everyday practice [72]. Additionally, since climate change is a cross-sectoral phenomenon [73], the government needs to formulate a longer-term cross-sectoral planning mainstreaming approach for more effective climate change adaptation policy implementation. In addition to the knowledge level gap, the study has unearthed the lack of funding options, which might present challenges to the effectiveness of adoption actions in forest ecosystem services. Climate change adaptation actions require sustainable funding mechanisms [74].

Finally, we noted some limitations in our study. For example, we employed an online survey approach in which we purposively chose forestry and climate change experts to assess the effectiveness, adaptation actions, and challenges of the NPCC in forest ecosystem services. As such, the results are limited to implementing the NPCC in the unique forest ecosystem services. These limitations restrict the applicability of these results and replicating them to other policies.

## 5. Conclusions

This paper assessed the adaptation actions supported by the NPCC in forest ecosystem services of Namibia. The paper focused on the effectiveness and challenges of adaptation actions to climate change. The results suggest that there have been improvements in the adaptation actions after introducing the policy in 2011. After the NPCC, higher effectiveness scores were noted in forest ecosystems, such as biodiversity, carbon sequestration, soil conservation, and socio-economic benefits. Biodiversity and carbon sequestration were significantly effective after the introduction of the policy.

Our results further revealed that the most significant challenges were the lack of awareness, which showed prominence before and after the policy's introduction. Afforestation, reforestation, awareness, and forestry research need strengthening to improve the effectiveness of the policy. In response to the challenges, the experts expressed that promoting awareness was the most critical improvement needed to improve the effectiveness of the NPCC. Although our results showed that adaptation actions supported by the NPCC were generally effective after the policy was introduced, some areas concerning policy implementation still need strengthening through research to help in sound decision-making.

The need for research on forest ecosystem services-based adaptation cannot be understated. Research involves testing, refining, and up-scaling adaptation actions to climate change approaches, policies, and legislation based on the local context. Therefore, we propose that future research should analyze the strengths, weaknesses, threats, and opportunities (SWOT) of the NPCC and consequently design/propose a framework for forest ecosystem services-based adaptation actions in the policy to improve adaptation actions.

**Author Contributions:** Conceptualization, A.N. (Andreas Nikodemus), M.H. and A.N. (Albertina Ndeinoma); methodology, A.N. (Andreas Nikodemus); software, A.N. (Andreas Nikodemus) and R.C.P.; validation, M.H.; formal analysis, A.N. (Andreas Nikodemus) and R.C.P.; investigation, A.N. (Andreas Nikodemus), M.H. and A.N. (Albertina Ndeinoma); resources, A.N. (Andreas Nikodemus) and M.H.; data curation, A.N. (Andreas Nikodemus), and A.N. (Albertina Ndeinoma); writing—original draft preparation, A.N. (Andreas Nikodemus); writing—review and editing, A.N. (Andreas Nikodemus); visualization, A.N. (Andreas Nikodemus); supervision, M.H.; project administration, M.H.; funding acquisition, M.H. All authors have read and agreed to the published version of the manuscript.

**Funding:** This work was supported by the Operational Program Research, Development, and Education, the Ministry of Education, Youth, and Sports of the Czech Republic grant no. CZ.02.1.01/0.0/0.0/16_019/0000803.

**Institutional Review Board Statement:** The study was conducted in accordance with the ethical conduct of the National Commission on Research Science and Technology (NCRST), approved on 19 October 2022, and the ethical clearance by the Ministry of Environment, Forestry and Tourism (MEFT) of Namibia, and approved by the Directorate of Forestry (DoF), dated 18 August 2022.

**Informed Consent Statement:** Written informed consent has been obtained from the Ministry of Environment, Forestry, and Tourism of Namibia to publish this paper.

**Data Availability Statement:** All data relevant to the study are included in the article.

**Acknowledgments:** We want to thank individual experts in climate change, policy instruments, and environmental sciences representing public, private, and academic institutions who participated in this study. Finally, we are grateful to the anonymous reviewers for their constructive reviews.

**Conflicts of Interest:** The authors declare that they have no known competing financial interests or personal relationships that could have appeared to influence the work reported in this paper.

## Appendix A

*Survey Questionnaire*

### National Policy on Climate Change for Namibia: Are there adequate actions for adaption in forest ecosystem services?

#### SECTION A: DEMOGRAPHIC INFORMATION

*INSTRUCTION: Please put a cross (x) against the appropriate box to specify your choice.*

**1. Institution represented**

Public ☐
Private ☐
Developmental partners/projects ☐
Academic/research ☐

**2. Experience in the field of forest ecosystem services/environmental science**

Less than 1 year ☐
1-5 years ☐
6-10 years ☐
10+ years ☐

**3. What is your position in the institution?**

Top management ☐
Middle management ☐
Lower management ☐
Junior/general employee ☐
Intern ☐
Others (specify) ……………………

**SECTION B: IMPLEMENTATION OF NPCC IN ADAPTATION IN FORESTRY**

1. How would you rate the following ecosystem services in terms of effective implementation of the instruments that were implemented between 2001 and 2010 before NPCC?

| STRATEGIC ACTIONS | RATING | | | | | | |
|---|---|---|---|---|---|---|---|
| | Extremely ineffective | Very ineffective | Ineffective | Not sure | Effective | Very effective | Extremely effective |
| Biodiversity | | | | | | | |
| Carbon sequestration | | | | | | | |
| Soil conservation | | | | | | | |
| Socio-economic benefits for communities | | | | | | | |
| Recreational and cultural values | | | | | | | |
| Watershed services | | | | | | | |
| High conservation values | | | | | | | |

2. Which instruments played the most significant role in governing actions for climate change adaptation in the context of forest ecosystem services before the NPCC implementation (2001-2010)?

| RATE | YEARS | | | | |
|---|---|---|---|---|---|
| | 2001-2002 | 2003-2004 | 2005-2006 | 2007-2008 | 2009-2010 |
| National Forestry Policy | | | | | |
| Forest Act | | | | | |
| National Environmental Education and Education for Sustainable Development Policy | | | | | |
| Communal Land Reform Act | | | | | |
| The Nature Conservation Ordinance No. 4 of 1975 | | | | | |
| Other (specify) | | | | | |

3. What were the main challenges for policy instruments for climate change adaptation before the implementation of NPCC?

Lack of awareness about climate change    ☐

High demands for land by local communities    ☐

Limited research in forestry and climate change    ☐

Adverse weather conditions    ☐

High rates of illegal logging    ☐

Poverty in rural areas    ☐

Lack of funding options    ☐

Other (specify) …………………………………

4. **Which of the following indicators for climate change adaptation were evident before the implementation of NPCC in the context of forest ecosystem services from 2001-2010?**

| RATE | YEARS | | | | |
|---|---|---|---|---|---|
| | 2001-2002 | 2003-2004 | 2005-2006 | 2007-2008 | 2009-2010 |
| Plantations and orchards establishment | | | | | |
| Improved awareness creation | | | | | |
| Improved law enforcement | | | | | |
| Reduced reliance of local communities on forest resources | | | | | |
| Improved funding options for adaptation | | | | | |
| Improved forestry research and innovation | | | | | |
| Improved conservation of ecosystems critically threatened by climate change | | | | | |
| Other (please specify) …………… | | | | | |

5. **How do you rate the following strategic actions for climate change adaptation in the context of forest ecosystem services from 2001-2010?**

| STRATEGIC ACTIONS | RATING | | | | | | |
|---|---|---|---|---|---|---|---|
| | Extremely ineffective | Very ineffective | Ineffective | Not sure | Effective | Very effective | Extremely effective |
| Afforestation and reforestation | | | | | | | |
| Law enforcement | | | | | | | |
| Altering local communities' reliance on forest resources | | | | | | | |
| Funding adaptation activities | | | | | | | |
| Forestry research | | | | | | | |
| Conservation of ecosystem services critically threatened by climate change | | | | | | | |
| Other (specify)………….. | | | | | | | |

6. **How would you rate your knowledge of the NPCC and its implementation strategies in forestry?**

Very good ☐
Good ☐
Familiar ☐
Neutral ☐
Poor ☐
Very poor ☐

7. **How would you rate the following ecosystem services in terms of effective implementation of the NPCC over the past nine years (2012-2021) of its implementation?**

| STRATEGIC ACTIONS | RATING | | | | | | |
|---|---|---|---|---|---|---|---|
| | Extremely ineffective | Very ineffective | Ineffective | Not sure | Effective | Very effective | Extremely effective |
| Biodiversity | | | | | | | |
| Carbon sequestration | | | | | | | |
| Soil conservation | | | | | | | |
| Socio-economic benefits for communities | | | | | | | |
| Recreational and cultural values | | | | | | | |
| Watershed services | | | | | | | |
| High conservation values | | | | | | | |

8. **Which of the following adaptation strategic actions of NPCC does your institution support/focus on the most? (You can choose as many according to the activities of your institution).**

Promoting afforestation and reforestation ☐

Supporting law enforcement ☐

Altering local communities' reliance on forest resources ☐

Providing funding for adaptation activities ☐

Supporting existing forestry research ☐

Conservation of forest ecosystems services critically threatened by climate change ☐

Other (specify) ………………………………………………………………………

9. **Which indicators for climate change adaptation have been evident during the implementation of NPCC in the context of forest ecosystem services?**

| RATE | YEARS | | | | |
|---|---|---|---|---|---|
| | 2012-2013 | 2014-2015 | 2016-2017 | 2018-2019 | 2020-2021 |
| Plantations and orchards establishment | | | | | |
| Improved awareness creation | | | | | |
| Improved law enforcement | | | | | |
| Reduced reliance of local communities on forest resources | | | | | |
| Improved funding options for adaptation | | | | | |
| Improved forestry research and innovation | | | | | |
| Improved conservation of ecosystems critically threatened by climate change | | | | | |
| Other (please specify) …………… | | | | | |

10. **How do you rate the following strategic actions for NPCC for climate change adaptation in the context of forest ecosystem services?**

| STRATEGIC ACTIONS | RATING | | | | | | |
|---|---|---|---|---|---|---|---|
| | Extremely ineffective | Very ineffective | Ineffective | Not sure | Effective | Very effective | Extremely effective |
| Afforestation and reforestation | | | | | | | |
| Law enforcement | | | | | | | |
| Altering local communities' reliance on forest resources | | | | | | | |
| Funding adaptation activities | | | | | | | |
| Forestry research | | | | | | | |
| Conservation of ecosystem services critically threatened by climate change | | | | | | | |
| Other (specify)…………... | | | | | | | |

11. **What is the main challenge facing the implementation of NPCC in climate change adaptation in the context of forest ecosystem services? (You can choose as many according to the activities of your institution).**

Lack of awareness about climate change ☐

High demands for land by local communities ☐

Limited research in forestry and climate change ☐

Adverse weather conditions ☐

High rates of illegal logging ☐

Poverty in rural areas ☐

Lack of funding options ☐

Other (specify) ……………………………………

12. **Which of the following would you describe as the most significant in terms of the strengths, weaknesses, opportunities, and threats of NPCC in promoting adaptation in forest ecosystem services?**

| Strengths | | Opportunities | |
|---|---|---|---|
| Understanding climate change | ☐ | Improve public awareness | ☐ |
| Good forestry management | ☐ | Various fundings options | ☐ |
| Good collaboration among stakeholders | ☐ | Support from NGOs | ☐ |
| Adequate adaptation science and technologies | ☐ | Urban agriculture | ☐ |
| Sound land use planning | ☐ | Agroforestry practices | ☐ |
| Sufficient funds | ☐ | Other (specify) | |
| Sufficient skills in tree planting | ☐ | | |
| Established protected areas | ☐ | | |
| Other (specify) | | | |
| **Weaknesses** | | **Threats** | |
| Poor understanding of climate change | ☐ | Deforestation and land degradation | ☐ |
| Poor forestry management | ☐ | Soil depletion | ☐ |
| Poor collaboration among stakeholders | ☐ | High rates of illegal logging | ☐ |
| Inadequate adaptation science and technologies | ☐ | Adverse weather conditions | ☐ |
| Poor land use planning | ☐ | Increased demands for land use | ☐ |
| Insufficient funds | ☐ | Poverty in rural areas | ☐ |
| Poor experience in tree planting | ☐ | Other (specify) | |
| Lack of established protected areas | ☐ | | |
| Other (specify) | | | |

13. **What recommendations would you make to improve the implementation of NPCC for climate change adaptation measures in the context of forest ecosystem services?**

.................................................................................................................................

.................................................................................................................................

## Appendix B

*Results*

**Table A1.** Effectiveness scores of the implemented adaptation actions by forest ecosystem services before and after NPCC.

| Forest Ecosystem Services | Before NPCC (N = 36) | After NPCC (N = 36) | *p*-Value |
|---|---|---|---|
| Biodiversity | 3.11 ± 0.92 | 4.36 ± 1.52 | <0.001 |
| Carbon sequestration | 2.75 ± 1.34 | 3.06 ± 1.35 | 0.338 |
| Soil conservation | 3.08 ± 1.32 | 3.39 ± 1.29 | 0.324 |
| Socio-economic benefits | 3.25 ± 1.20 | 3.44 ± 1.34 | 0.519 |
| Recreational and cultural values | 3.72 ± 1.09 | 3.39 ± 1.34 | 0.249 |
| Watershed services | 3.31 ± 0.89 | 3.17 ± 1.23 | 0.585 |
| High conservation values | 3.56 ± 1.40 | 3.14 ± 1.25 | 0.187 |

Data are presented as mean ± standard deviation (sd); independent *t*-test was applied to compare mean scores before and after NPCC implementation. Significantly different at $p < 0.05$.

**Table A2.** Indicators of actions scores of the implemented policy before and after NPCC.

| Adaptation Action Indicators | Before NPCC (N = 36) | After NPCC (N = 36) | *p*-Value |
|---|---|---|---|
| Afforestation and reforestation | 3.17 ± 1.56 | 3.33 ± 1.69 | 0.665 |
| Law enforcement | 3.33 ± 1.51 | 3.61 ± 1.63 | 0.455 |
| Altering local communities' reliance on forest resources | 3.42 ± 1.44 | 3.56 ± 1.59 | 0.699 |
| Funding adaptation activities | 3.11 ± 1.58 | 3.53 ± 1.42 | 0.244 |
| Forestry research | 3.17 ± 1.63 | 3.28 ± 1.60 | 0.771 |
| Conservation of ecosystem services critically threatened by climate change | 30.6 ± 1.41 | 3.53 ± 1.42 | 0.162 |
| Stakeholders' collaboration | 2.94 ± 1.64 | 3.53 ± 1.40 | 0.109 |

Data are presented as mean ± sd; independent *t*-test was applied to compare mean scores before and after NPCC implementation. Significantly different at $p < 0.05$.

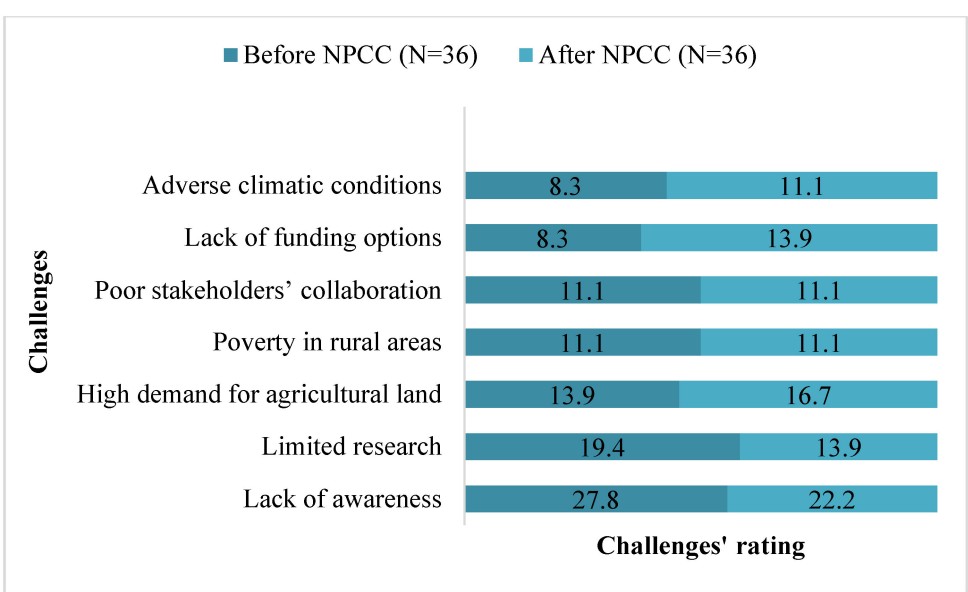

**Figure A1.** Challenges in the implementation of adaptation actions before and after the NPCC. Challenges' ratings are presented as %.

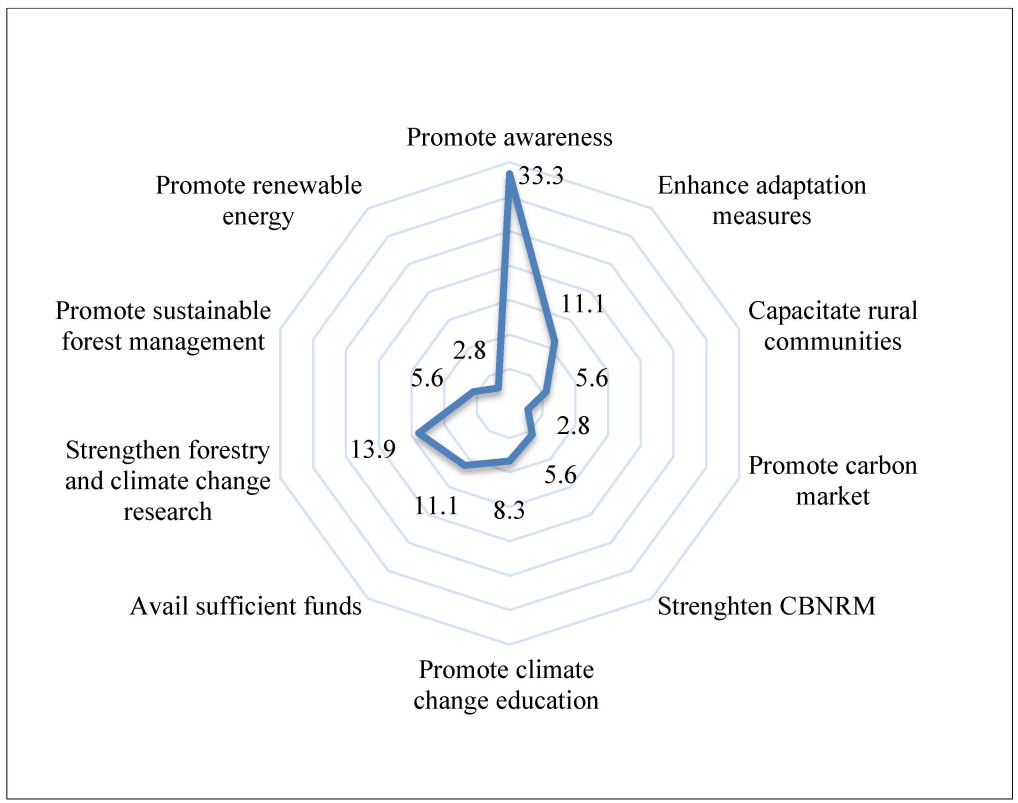

**Figure A2.** Proposed improvements in the implementation of the NPCC, according to the experts. Proposed improvements for the NPCC are presented as %.

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
