# Peer review of "Forest Ecosystem Services-Based Adaptation Actions Supported by the National Policy on Climate Change for Namibia: Effectiveness, Indicators, and Challenges"

_forests, doi:10.3390/f13111965_

Round 1
Reviewer 1 Report
Dear authors!
The paper is devoted to the actual topic of studying ecosystem services. This is an extremely interesting and relevant topic. However, the article has a number of shortcomings. In the future, the level of the paper can be significantly improved.
General comments on the paper are as follows:
1. The analysis of the world experience in the study of the topic “Forest ecosystem services” will significantly improve the article. Specify what documents exist in other countries of the world, on other continents - not only in Africa? Is it possible to use their experience in Namibia? It is necessary to compare how they are worse or better than NPCC?
2. More field research needs to be done. Indicate in the work how many respondents were sent questionnaires and how many questionnaires were received back with the results of the questionnaires? It is necessary to cover the broader sections of society with the questionnaire. Indicate whether the questionnaires were sent to ordinary citizens, not to experts?
3. Expand the "Discussion" section. Compare with other programs. State why your results are better than those of other researchers?
4. Justify the expediency of highlighting the section “5. Study limitations". It is not clear why it should be separated from the "Discussion" section?
5. The paper needs to be radically revised and it can be accepted.
Author Response
Dear reviewer 1,
Please see the attachment for our response to your comments.
Kind regards,
Andreas Nikodemus

Reviewer 2 Report
Dear Authors,
Congratulations on your manuscript entitled "Forest ecosystem services-based adaptation actions supported by the National Policy on Climate Change for Namibia: Effectiveness, indicators, and challenges". I have reviewed it and think it is suitable for publication in Forests journal after minor corrections.
The most important note I make is about the background. I have noticed that the bibliographic base of your manuscript is not as broad as you would expect from a well designed manuscript as you have prepared it: only 49 references to support the basis of your work. I've noticed that you repeat some references and it's not wrong, but why don't you implement your research background by adding more references, especially (but not limited to) in the Introduction section? I have taken the liberty of suggesting some that I believe may be useful to you, if you deem it appropriate. Later you can discuss them more fully in the Discussion section.
Please check the different notes I have indicated in the manuscript.
Finally, in order to increase the visibility of your paper I recommend changing the keywords I deleted. If you change them by other keywords, you will increase the probability that your paper could be found by future readers when they look for it in some databases, like Scopus for example. If you repeat the same words in the article title and in keywords, less people could find your work. So, you must think about the visibility of your research. For example, in Scopus, when you look for a specific word normally in the same box you look for “Article title, Abstract and Keywords”, so, if you have different ones in all of them, obviously the visibility of your manuscript will increase.
Best wishes.

Author Response
Dear Reviewer 2,
Thank you for your constructive comments and advice. Kindly see the responses in the attachment.
Kind regards,
Andreas Nikodemus

Round 2
Reviewer 1 Report
This is a good revision, thank you.
The article can be published.